# Rare de novo damaging DNA variants are enriched in attention-deficit/hyperactivity disorder and implicate risk genes

Emily Olfson [1,2] ✉, Luis C. Farhat[1,3], Wenzhong Liu[1], Lawrence A. Vitulano[1], Gwyneth Zai[4,5], Monicke O. Lima[3], Justin Parent [6,7,8], Guilherme V. Polanczyk [3], Carolina Cappi [9], James L. Kennedy [4,5] & Thomas V. Fernandez [1,10] ✉

Research demonstrates the important role of genetic factors in attention-deficit/hyperactivity disorder (ADHD). DNA sequencing of families provides a powerful approach for identifying de novo (spontaneous) variants, leading to the discovery of hundreds of clinically informative risk genes for other childhood neurodevelopmental disorders. This approach has yet to be extensively leveraged in ADHD. We conduct whole-exome DNA sequencing in 152 families, comprising a child with ADHD and both biological parents, and demonstrate a significant enrichment of rare and ultra-rare de novo gene-damaging mutations in ADHD cases compared to unaffected controls. Combining these results with a large independent case-control DNA sequencing cohort (3206 ADHD cases and 5002 controls), we identify *lysine demethylase 5B (KDM5B)* as a high-confidence risk gene for ADHD and estimate that 1057 genes contribute to ADHD risk. Using our list of genes harboring ultra-rare de novo damaging variants, we show that these genes overlap with previously reported risk genes for other neuropsychiatric conditions and are enriched in several canonical biological pathways, suggesting early neurodevelopmental underpinnings of ADHD. This work provides insight into the biology of ADHD and demonstrates the discovery potential of DNA sequencing in larger parent-child trio cohorts.

Attention-deficit/hyperactivity disorder (ADHD) affects ~3–5% of children worldwide[1] and places a significant burden on individuals, their families, and the community[2]. ADHD is highly heritable (~70–80%)[3], so identifying risk genes will increase our understanding of underlying biological processes. Recent case–control genome-wide studies have identified ADHD risk loci by assessing common single-nucleotide

polymorphisms (SNPs) through genome-wide association studies (GWAS)[4,5]. However, to date, SNP-heritability has only accounted for a small portion (~15–30%) of overall heritability estimates, suggesting that other genetic factors, including rare genetic variants, may play an important role in ADHD risk[6]. Indeed, previous studies have demonstrated that rare copy number variants[7] and very rare protein-

[1]Child Study Center, Yale University, New Haven, CT, USA. [2]Wu Tsai Institute, Yale University, New Haven, CT, USA. [3]Division of Child & Adolescent Psychiatry, Department of Psychiatry, Faculdade de Medicina FMUSP, Universidade de São Paulo, São Paulo, Brazil. [4]Tanenbaum Centre, Molecular Brain Sciences Department, Campbell Family Mental Health Research Institute, Centre for Addiction and Mental Health, Toronto, ON, Canada. [5]Institute of Medical Science and Department of Psychiatry, University of Toronto, Toronto, ON, Canada. [6]University of Rhode Island, Kingston, RI, USA. [7]Bradley/Hasbro Children's Research Center, E.P. Bradley Hospital, Providence, RI, USA. [8]Alpert Medical School of Brown University, Providence, RI, USA. [9]Department of Psychiatry at Icahn School of Medicine at Mount Sinai Hospital, New York, NY, USA. [10]Department of Psychiatry, Yale University, New Haven, CT, USA. ✉e-mail: emily.olfson@yale.edu; thomas.fernandez@yale.edu

truncating variants in evolutionarily constrained genes[8] are enriched in ADHD. Despite previous research considering these different categories of genetic variation, no specific high-confidence risk genes have yet been identified for ADHD.

Detecting rare de novo genetic variants using parent–child trios has proven to be a powerful approach for risk gene discovery in other neurodevelopmental disorders such as autism spectrum disorder (ASD)[9], developmental delay/intellectual disability[10], and Tourette's disorder[11], leading to the identification of a plethora of risk genes. Since the background rate of de novo variants in the population is low, finding an elevated rate of damaging de novo variants suggests that we can leverage these variants to identify risk genes and underlying biological pathways. Studies examining de novo copy number variants (CNVs) indicate a greater rate of these variants in ADHD cases compared to controls[12,13], but given the large number of genes disrupted by CNVs, it is challenging to identify specific risk genes from these variants. Whole-exome DNA sequencing studies enable the identification of de novo sequence variants affecting single genes. A few small DNA sequencing studies of parent–child trios with ADHD[14–17] have identified rare de novo sequence variants, supporting the discovery potential of applying this approach in larger ADHD cohorts.

Here, we conduct whole-exome DNA sequencing in 152 parent–child trios (456 individuals in total), comprising a child with ADHD and both biological parents, and demonstrate that rare and ultra-rare de novo protein-truncating and damaging missense variants are enriched in ADHD cases compared to unaffected controls. Combining our results with a large independent case–control DNA sequencing study (3206 ADHD cases and 5002 typically developing controls)[8], we identify *lysine demethylase 5B* (*KDM5B*) as a high-confidence risk gene for ADHD and identify three other potential risk genes. Finally, we show overlap among genes harboring de novo damaging variants in ADHD with previously reported risk genes for

other psychiatric conditions, and we conduct exploratory analyses to identify biological pathway enrichment. These findings provide a critical step forward toward improving our etiologic understanding of ADHD, which may, in the future, inform the treatment of this common and impairing condition.

## Results

### Rare and ultra-rare de novo damaging variants are enriched in ADHD probands

We performed whole-exome DNA sequencing in 152 parent–child trios with ADHD collected from four sites (Supplementary Data 1). We pooled this sequencing data and performed joint variant calling with whole-exome sequencing from 788 parent–child trios without ADHD, already sequenced as part of the Simons Simplex Collection. After applying our quality control methods, we compared rates of de novo variants in 147 ADHD parent–child trios and 780 control parent–child trios. Based on studies of other childhood-onset neuropsychiatric conditions[11,18–20], we expected to find a greater rate of rare de novo damaging variants in ADHD probands versus controls. Damaging variants include protein-truncating variants (PTVs, including premature stop codons, frameshift, and splice site variants) and missense variants predicted to be damaging (Mis-D) by a "missense badness, PolyPhen-2, constraint" (MPC) score > 2[21].

Results from this burden analysis demonstrate a greater rate of both rare and ultra-rare de novo damaging variants (PTVs + Mis-D) in ADHD cases versus unaffected controls (Fig. 1, Supplementary Table 1, Supplementary Data 2). For rare de novo damaging variants (non-neuro gnomAD allele frequency < 0.001), the rate ratio was 1.67 (95% CI 1.08–2.53, p = 0.03). We found a greater difference between cases and controls when narrowing our analysis to ultra-rare de novo damaging variants (non-neuro gnomAD allele frequency < 0.00005), with a rate ratio of 1.93 (95% CI 1.24–2.97, p = 0.007) (Fig. 1,

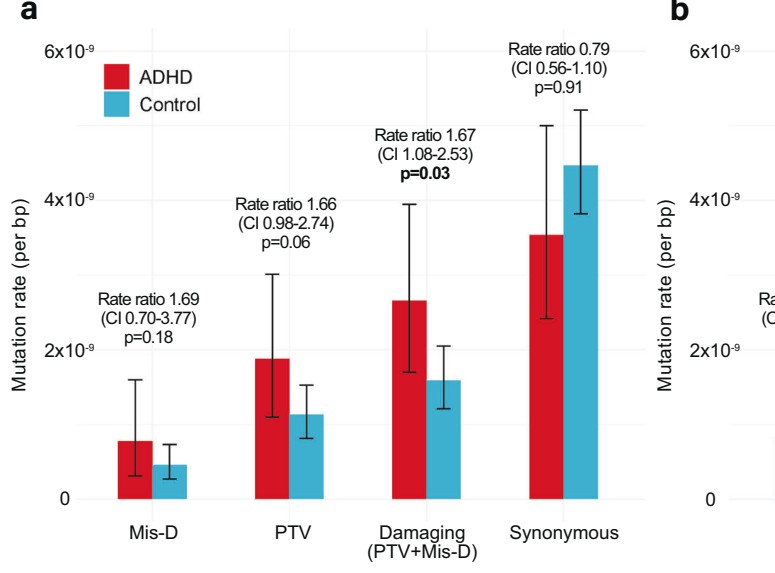

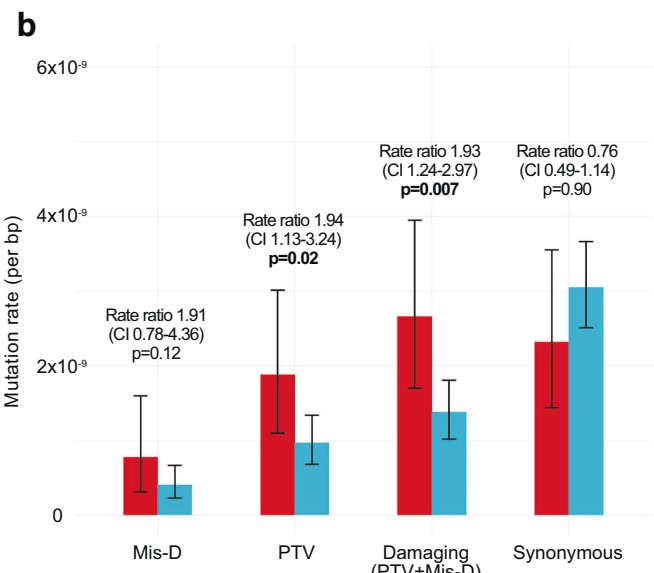

**Fig. 1 | De novo mutation rates in ADHD versus control subjects.** Rates of **a** rare and **b** ultra-rare de novo damaging mutations are enriched in ADHD probands (n = 147) compared to controls (n = 780). Rare variants have an allele frequency <0.001 (0.1%) in the non-neuro subset of the Genome Aggregation Database (gnomAD), and ultra-rare de novo variants have an allele frequency of <0.00005 (0.005%) in the non-neuro subset of gnomAD. For each variant class, the mutation rate was calculated by dividing the total number of mutations in that class by the "callable" loci across all families meeting sequencing depth and quality score thresholds. Mutation rates were calculated separately for ADHD cases (red) and

controls (blue). Error bars show 95% confidence intervals for mutation rates calculated based on a Poisson distribution. Mutation rates between cases and controls were compared with a one-tailed rate ratio test with a significance threshold of p < 0.05. Bold p-values are significant. Values are provided in Supplementary Table 1 and Supplementary Data 2. PTVs, protein-truncating variants, including frameshift, splice site, and stop-gain variants. Mis-D, missense variants predicted to be damaging with "missense badness, PolypPhen-2 constraint" (MPC) score > 2. Source data are provided as a Source Data file.

Supplementary Table 1). Within the subset of ultra-rare de novo damaging variants, we found a greater rate of PTVs (rate ratio 1.94, 95% CI 1.13–3.24, $p = 0.02$) and a trend towards an increased rate of Mis-D variants in cases versus controls (rate ratio 1.91, 95% CI 0.78–4.36, $p = 0.12$). As anticipated, we did not find differences in rates of de novo variants between cases and controls when including all (damaging and non-damaging) rare or all ultra-rare variants (Supplementary Table 1).

### Recurrent ultra-rare damaging variants identify ADHD risk genes

Among 147 ADHD parent–child trios passing quality control, we identified 24 ultra-rare de novo damaging variants in 23 individuals (Table 1, Supplementary Data 2). Among 780 control parent–child trios passing quality control, we identified 51 ultra-rare de novo damaging variants in 50 individuals (Supplementary Data 2). The list of genes harboring rare or ultra-rare de novo damaging variants in ADHD cases did not overlap with genes harboring rare or ultra-rare de novo damaging variants in control parent–child trios passing quality control (Supplementary Data 2). One gene, *KDM5B*, had two de novo PTVs in unrelated individuals in our ADHD trio cohort. To identify ADHD risk genes (genes harboring damaging variants more often than expected by chance), we combined our de novo parent–child trio findings with counts of ultra-rare PTVs and Mis-D (MPC > 2) variants identified in a large independent ADHD case–control dataset (3206 ADHD cases and 5002 typically developing controls)[8]. Using this combined dataset, we applied the Transmission And De novo Association test (extTADA)[22]. We identified *KDM5B* as a high-confidence risk gene (FDR = 0.04) and three potential risk genes for ADHD: *YLPM1* (FDR = 0.20), *CTNND2* (FDR = 0.26), and *GNB2L1* (FDR = 0.30) (Fig. 2, Supplementary Data 3). This extTADA analysis estimates that 1057 genes (95% CI 219–2791) contribute to ADHD risk.

### Genes harboring de novo damaging variants in ADHD overlap with risk genes for other psychiatric conditions

Using the list of 23 genes with ultra-rare de novo damaging variants (PTV and Mis-D) in 147 ADHD probands (Table 1, Supplementary Data 2), we identified overlap with risk genes for other conditions. Six of these 23 genes were recently reported as likely risk genes (FDR < 0.05) for neurodevelopmental disorders (NDD) in the largest meta-analysis of ASD and developmental delay[9], including *FBXO11* (FDR = 0), *KDM5B* (FDR = 0), *STAG1* (FDR = 1.98 × 10^−7), *CTNNA2* (FDR = 9.49 × 10^−5), *EML6* (FDR = 0.002), and *PAK1* (FDR = 0.006) (Table 1, Supplementary Data 4). Using the Gene4Denovo database[23], *KDM5B* is also a risk gene for ASD (FDR = 0), undiagnosed developmental disorders (FDR = 0), congenital heart disease (FDR = 0.005), and across all disorders in the Gene4Denovo database (FDR = 0). *FBXO11* and *STAG1* were also both associated with undiagnosed developmental disorders (FDR = 0 and 0.00002, respectively) and across all disorders (FDR = 0 for both), and *PAK1* was a risk gene across all disorders (FDR = 0.009) (Supplementary Data 4). Additionally, we identified an overlap between our list of 23 genes with ultra-rare de novo damaging variants in ADHD probands and gene-mapped loci from common variant GWAS studies in neuropsychiatric disorders in the GWAS Catalog (Supplementary Data 5).

### Exploratory gene ontology and pathway enrichment

Using this same list of 23 genes harboring ultra-rare de novo damaging variants in ADHD trios, we also conducted exploratory analyses to identify enriched gene ontology and biological pathways. Several gene ontology and pathway-based sets were enriched for these 23 genes identified in ADHD (Supplementary Data 6). The top pathway-based sets were CXCR4-mediated signaling events ($q = 0.004$), Sema3A PAK-dependent Axon repulsion ($q = 0.004$), and ectoderm differentiation ($q = 0.009$).

## Discussion

In this largest parent–child trio whole-exome DNA sequencing study of ADHD to date, we found a significantly greater rate of rare and ultra-rare de novo damaging variants in children with ADHD compared to unaffected controls (Fig. 1). Combining our trio sequencing data with results from a large independent case–control DNA sequencing dataset, we identified *KDM5B* as a high-confidence risk gene for ADHD and three other potential risk genes, *YLPM1, CTNND2, and GNB2L1* (Fig. 2).

Our sequencing data identified a 1.67-fold enrichment of rare de novo damaging variants in ADHD cases compared to unaffected controls, and this enrichment was greater (1.93-fold) when narrowing to ultra-rare de novo damaging variants (Fig. 1, Supplementary Table 1). It is important to note that these estimated enrichments have wide confidence intervals, so caution is warranted in interpreting these results, and replication in larger ADHD parent–child trio cohorts is needed. Nevertheless, our observed mutation rates and our enrichment of rare de novo PTV and Mis-D variants are of a similar magnitude to those reported in other parent-offspring trio studies examining de novo variation in other neurodevelopmental disorders, including ASD and Tourette's disorder[11,20]. Our enrichment of rare and ultra-rare de novo damaging variants in ADHD cases compared to controls is also consistent with findings from the largest case–control DNA sequencing study examining rare variations in ADHD. This study also reported enrichment of ultra-rare PTVs in constrained genes in ADHD cases, and the rate of these variants in ADHD was comparable to ASD cases[8]. Similar to ASD[20], we found that PTVs comprise a greater proportion of rare de novo variants than transmitted variants in ADHD (Supplementary Data 2). Our finding of enriched rare de novo damaging variants in ADHD adds information about the genomic architecture of the disorder and supports the value of DNA sequencing studies in larger ADHD parent–child trio cohorts to identify risk genes in a manner that has led to the identification of over 100 high-confidence risk genes in ASD. Similarly, our study suggests at least hundreds of genes contributing to ADHD risk, highlighting an efficient path toward systematic risk gene discovery in ADHD.

Our study identified ultra-rare de novo PTV variants in *KDM5B* in two unrelated individuals with ADHD (Table 1, Supplementary Data 2). *KDM5B* is a histone-modifying enzyme that specifically removes methyl groups from lysine 4 on histone 3 (H3K4 demethylase), leading to epigenetic regulation of gene expression. The gene is often studied in association with cancer, but rare damaging variants in *KDM5B* have been more recently associated with various other conditions and functions, including congenital heart disease, embryonic development, muscle strength, DNA repair, primary complex motor stereotypies in children, cognitive functioning in adults, ASD, and developmental disorders more broadly[10,20,24–31]. In individuals with an intellectual disability or developmental delay, *KDM5B* mutations often follow a recessive inheritance pattern with homozygous or compound heterozygous mutations[32,33]. Heterozygous damaging mutations have been reported in the Deciphering Developmental Disorders Study[34] and in ASD probands[20,30]. However, individuals with ADHD in our study harboring ultra-rare de novo PTV variants in *KDM5B* did not have diagnoses of ASD or intellectual disability. Consistent with our findings, evidence suggests considerable pleiotropy and gene dosage effects associated with this gene, in contrast to most neurodevelopmental disorder risk genes[28,29]. Our findings add to this evidence and suggest that ADHD is included in the spectrum of phenotypic changes that may occur in the context of rare damaging variants in *KDM5B*.

In addition, we identified individuals with ADHD with de novo damaging variants in the genes *FBXO11, STAG1, and CTNNA2*. These genes have high constraint (pLI) scores and have been previously identified as high-confidence risk genes for neurodevelopmental disorders[9] (Table 1, Supplementary Data 2). *FBXO11* encodes an F-box protein, and de novo variants have been associated with syndromic intellectual disability and behavioral difficulties, including ADHD[35,36].

**Table 1 | Ultra-rare de novo damaging variants identified in ADHD probands**

| ID | Gene[a] | Protein change | Genomic coordinate | Ref. allele | Alt. allele | Variant class[b] | pLI score[c] | Non-neuro gnomAD AF | Reported as likely risk gene for NDD[d] |
|---|---|---|---|---|---|---|---|---|---|
| ADHD6,p1 | RBBP6 | p.Y170C | Chr16:24567213 | A | G | missense (MPC 2.1) | 1 | 0 | no |
| ADHD14,p1 | YLPM1 | p.R1646X | Chr14:75276497 | C | T | stopgain | <0.001 | 0 | no |
| ADHD25,p1 | SEC1SBP2L | p.T934fs | Chr15:49284809 | TG | T | frameshift deletion | 0.981 | 0 | no |
| ADHD33,p1 | NARS2 | p.R54X | Chr11:78282471 | G | A | stopgain | <0.001 | 3.36E-05 | no |
| ADHD37,p1 | CTNNA2 | p.R348W | Chr2:80808942 | C | T | missense (MPC 2.6) | 0.975 | 0 | yes (ASD) |
| ADHD44,p1 | BEST4 | p.L207fs | Chr1:45251759 | CAGAG | C | frameshift deletion | <0.001 | 0 | no |
| ADHD50,p1 | KDM5B | p.R1093X | Chr1:202704703 | G | A | stopgain | <0.001 | 1.12E-05 | yes (ASD and DD) |
| ADHD57,p1 | STAG1 | p.R1088X | Chr3:136068009 | G | A | stopgain | 1 | 0 | yes (DD) |
| ADHD58,p1 | KDM5B | p.D806fs | Chr1:202711840 | TC | T | frameshift deletion | <0.001 | 0 | yes (ASD and DD) |
| ADHD61,p1 | EML6 | p.R313X | Chr2:55071273 | C | T | stopgain | <0.001 | 0 | yes |
| ADHD69,p1 | CFL1 | p.Y82C | Chr11:65623472 | T | C | missense (MPC 3.2) | 0.990 | 0 | no |
| ADHD71,p1 | OCEL1 | p.G34X | Chr19:17337532 | G | T | stopgain | <0.001 | 0 | no |
| | TTC26 | p.G24D | Chr7:138819468 | G | A | splicing | <0.001 | 0 | no |
| ADHD84,p1 | PLD5 | p.D28fs | Chr1:242511463 | TG | T | frameshift deletion | <0.001 | 0 | no |
| ADHD86,p1 | FBXO11 | p.P918S | Chr2:48035289 | G | A | missense (MPC 2.6) | 1 | 0 | yes (ASD and DD) |
| ADHD95,p1 | CHST15 | p.L214fs | Chr10:125804342 | G | GGT | frameshift insertion | <0.001 | 0 | no |
| ADHD98,p1 | EHBP1L1 | p.M1429fs | Chr11:65359271 | CATGG | C | frameshift deletion | <0.001 | 0 | no |
| ADHD99,p1 | TUBB | p.G233E | Chr6:30691669 | G | A | missense (MPC 3.3) | 1 | 0 | no |
| ADHD107,p1 | CTNND2 | p.V229fs | Chr5:11236867 | AC | A | frameshift deletion | 1 | 0 | no |
| ADHD117,p1 | GOLGB1 | p.Y288X | Chr3:121435768 | A | C | stopgain | 0.003 | 0 | no |
| ADHD130,p1 | L1TD1 | p.S580X | Chr1:62676185 | C | A | stopgain | <0.001 | 0 | no |
| ADHD134,p1 | GNB2L1 | p.N244D | Chr5:180665146 | T | C | missense (MPC 2.2) | 1 | 0 | no |
| ADHD141,p1 | PAK1 | c.C44T | Chr11:77103522 | G | A | missense (MPC 2.8) | 0.059 | 0 | yes (DD) |
| ADHD144,p1 | USP54 | p.R58X | Chr10:75331247 | G | A | stopgain | <0.001 | 3.30E-05 | no |

Ref. allele reference allele, Alt. allele alternative allele, gnomAD the Genome Aggregation Database, AF allele frequency, MPC "missense badness, PolyPhen-2, constraint" scores, NDD neurodevelopmental disorder, ASD autism spectrum disorder, DD developmental delay.

[a]Genetic variants were annotated with ANNOVAR using RefGene hg19 definitions.

[b]Damaging de novo variants were defined as protein-truncating variants (frameshift insertions, frameshift deletions, stop codon change, canonical splice site variants) or missense variants predicted to be damaging by an MPC "missense badness, PolyPhen-2, constraint" score > 2. For missense variants, the MPC score of the variant is listed in parenthesis.

[c]pLI scores from gnomAD are provided for genes to predict tolerability to protein truncating variation with pLI scores greater than 0.9 thought to be extremely intolerant to truncating variants.

[d]Overlap with likely risk genes (false discovery rate, FDR <0.05) for NDD in largest meta-analysis of ASD (n = 63,237) and DD (n = 91,605)[9]. If FDR <0.05 for ASD and DD separately, it is also included in parentheses.

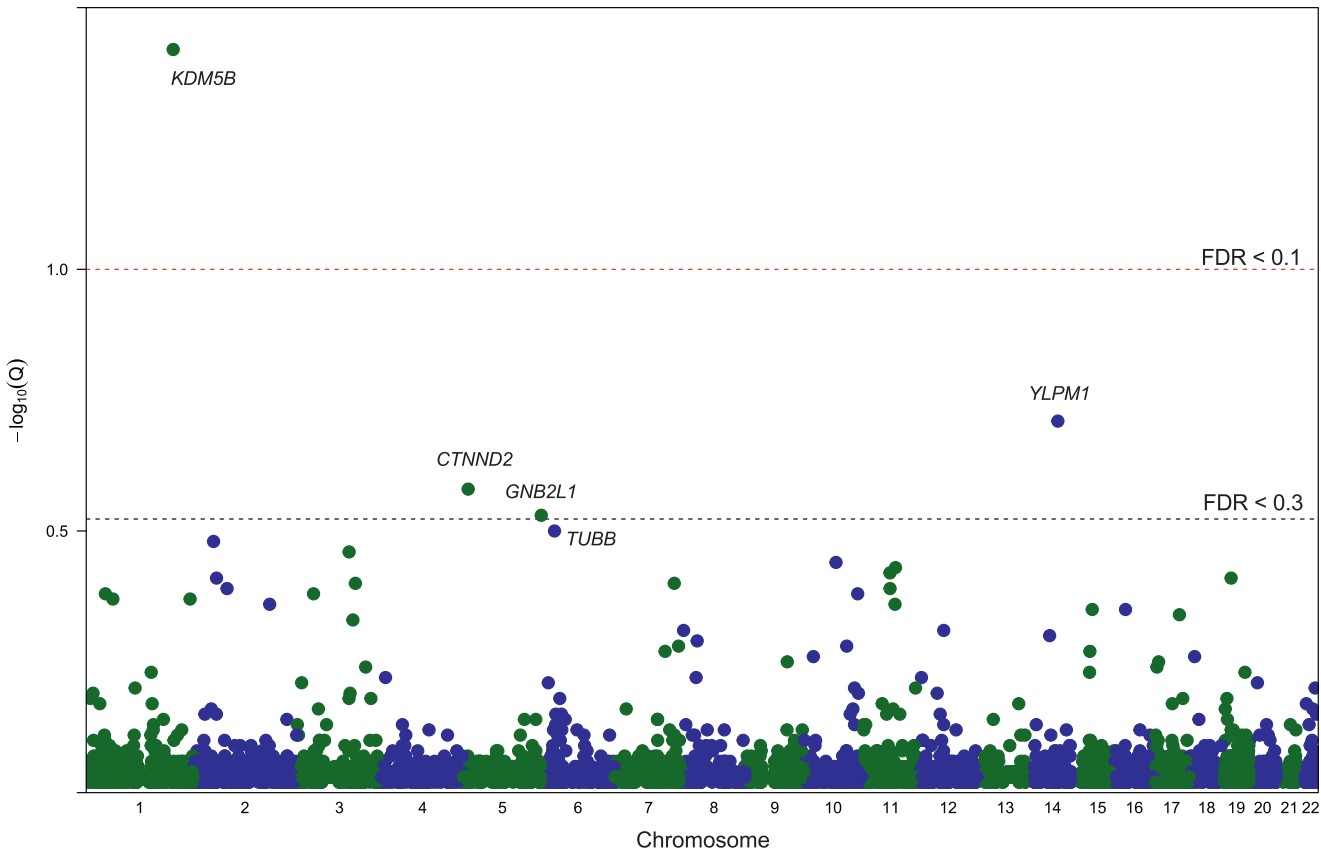

**Fig. 2 | Gene-based test results for ADHD, combining ultra-rare de novo damaging variants and independent case–control data.** Results from the extension of the Transmission And De novo Association test (extTADA) examining ultra-rare de novo protein-truncating variants and missense variants predicted to be damaging (MPC score > 2) from 147 ADHD parent–child trios and an independent group of 3206 ADHD cases and 5002 typically developing controls. Genes are organized by chromosome with the green dots indicating genes on odd-numbered chromosomes and the blue dots indicating genes on the even-numbered chromosomes. The top 5 gene symbols with the lowest *q* values are listed. Only one gene, *KDMSB*, is classified as a high-confidence risk gene (FDR, false discovery rate < 0.1). Three genes, *YLPM1* (FDR = 0.20), *CTNND2* (FDR = 0.26), and *GNB2L1* (FDR = 0.30) are classified as potential risk genes (FDR < 0.3). Values are provided in Supplementary Data. Source data are provided as a Source Data file.

*STAG1* encodes a component of cohesion involved with the separation of sister chromatids and has been associated with syndromic intellectual disability[37]. *CTNNA2* encodes a brain-expressed alpha-catenin protein involved with neuronal migration and synaptic plasticity[38], and SNPs within this gene and its regulatory region have been associated with impulsivity[39], excitement seeking[40], and perseverative negative thinking[41]. Our study identified ultra-rare damaging de novo variants in these genes in children with ADHD who did not have intellectual disability or other known genetic syndromes. We did not see damaging variants in *FBXO11* or *STAG1* in any controls (Table 1, Supplementary Data 2, Supplementary Data 3), while one out of 5002 control subjects from the large ADHD case–control dataset[8] was found to have a PTV in *CTNNA2* (Supplementary Data 3). This highlights the potential range of clinical manifestations that may occur due to de novo damaging variants in these genes and suggests potential clinical implications for identifying de novo damaging variants. Several additional genes with rare de novo damaging variants in ADHD probands are discussed in the Supplementary Discussion.

Genes harboring rare de novo gene-damaging variants in the ADHD cases not only overlapped with high-confidence risk genes identified in previous DNA sequencing studies of other neuropsychiatric conditions (Supplementary Data 4) but also overlapped with genes mapped from genome-wide significant common variants identified in previous GWA studies (Supplementary Data 5). Although there was no overlap with the 76 prioritized risk genes by positional and functional annotation or the 45 exome-wide significant genes identified in the recent large ADHD GWAS[4], there was overlap between genes mapped from externalizing-related disorders more broadly[42]. These findings add to the growing evidence supporting the convergence of common and rare variants in ADHD[4] and psychiatric disorders in general[20,43].

Finally, we conducted exploratory ontology and pathway analyses of genes harboring de novo damaging variants in our ADHD cases. In interpreting these results, it is important to note that many of these genes may not be true ADHD risk genes, and replication of these exploratory findings is needed as more high-confidence risk genes are identified. Nevertheless, we observed a significant enrichment of several biological processes. Of note, one of the top pathways is ectoderm differentiation (Supplementary Data 6), suggesting early neurodevelopmental underpinnings of ADHD. In the largest recent GWAS study of ADHD, gene-linked loci were enriched for expression in early brain development[4], also suggesting the possible role of early embryonic changes in the development of ADHD.

Aside from those already mentioned, this study has additional limitations that should be considered. For comparing mutation rates, the ideal controls would have been sequenced simultaneously as the cases and assessed for ADHD. This study prioritized sequencing ADHD parent–child trios and used controls previously sequenced as part of the Simons simplex collection (SSC) using similar methods and scored in the normal range of the ADHD subscale of the child behavioral checklist (CBCL) or the adult behavioral checklist (ABCL). By focusing on the intersection of the capture platforms, we tried to minimize

batch effects as done in other DNA sequencing studies[11,19,44]. It is important to note that these SSC control siblings may have an elevated genetic variant load compared to a population cohort; mutation rates and gene enrichments reported in our ADHD cases would have to overcome this potential elevation in controls to achieve statistical significance. However, these SSC control siblings are often used in parent–child trio studies, offering an advantage for future cross-disorder comparisons. Finally, our study focused on the coding region of the genome, and it is possible that relevant rare variants also occur in the noncoding region. Currently, understanding the biological and clinical relevance of non-coding variation remains challenging, but future studies of ADHD may utilize whole-genome sequencing technologies.

Despite these limitations, our study is important because it demonstrates enrichment of rare and ultra-rare de novo damaging variants in ADHD cases compared to unaffected controls and identifies *KDM5B* as a high-confidence risk gene for ADHD as well as other candidate risk genes for future study. These findings reinforce the value of DNA sequencing of parent–child trios in larger cohorts to identify additional risk genes for ADHD. Identifying risk genes that can be studied in model systems may offer further insight into the underlying biology of ADHD and can potentially inform clinical care for individuals and families.

## Methods

### Participants
This research complies with all relevant ethical regulations. This study protocol examining de-identified genetic data was reviewed by the Yale Institutional Review Board, Human Investigation Committee, and Human Subjects Committee, and determined not to be human subjects research (IRB Protocol ID 2000023609). This protocol did not include consent or compensation. Informed consent/assent was obtained at the time that the samples were collected from the participating sites. A total of 152 parent–child trios (456 individuals in total), comprising a child meeting DSM-IV or DSM-5 criteria for the diagnosis of ADHD and both biological parents, were included in this study. Trios were identified from four sites: the University of São Paulo School of Medicine ($n = 30$), the Center for Addiction and Mental Health in Toronto ($n = 37$), Florida International University ($n = 13$), and the Genizon biobank from Génome Québec ($n = 72$). All subjects were assessed by structured clinical interviews. The average age at evaluation was 4.9 years (s.d. 0.7 years) for the University of São Paulo School of Medicine site, 8.2 years (s.d. 1.9 years) for the Florida International University site, and 8.3 years (s.d. 1.6 years) for the Genizon biobank from Génome Québec site. Age data was unavailable for samples from the Center for Addiction and Mental Health in Toronto. Exclusion criteria included a diagnosis of ASD, intellectual disability, psychosis, mood disorders (including bipolar disorder), and clinically significant medical or neurological disease. No exclusions were made based on self-reported gender or biological sex, which was verified by DNA sequencing data (individual-level data in Supplementary Data 1 and 2). We prioritized the study of simplex (no known family history of ADHD) parent–child trios to increase the likelihood of detecting deleterious de novo variants. Control subjects were 788 unaffected parent–child trios, selected from the Simons Simplex Collection from the National Institutes of Health Data Archive (https://nda.nih.gov/edit_collection.html?id=2042)[45]. Control subjects did not have ASD and were selected to be in the normal range for the attentional problems subscale from the CBCL or the ABCL ($t$ score < 64.5), which predicts ADHD diagnosis[46].

### Whole-exome DNA sequencing
Exome capture and whole-exome DNA sequencing of DNA from 80 children with ADHD and their parents were conducted at the Yale Center for Genomic Analysis (YCGA) using the IDT xGen V1 capture and the Illumina NovaSeq6000 sequencing instrument. An additional 72 ADHD parent–child trios were sequenced by Genome Quebec using the Agilent SureSelect All Exon V7 capture and the Illumina NovaSeq6000 sequencing instrument. 788 control parent–child trios were previously sequenced as part of the Simons Simplex Collection, using the NimbleGen SeqCap EzExomeV2 capture and the Illumina HiSeq 2000 sequencing instrument. We performed joint variant calling with sequencing data from all cases and controls (940 trios, 2820 individuals in total).

### Sequencing alignment and variant identification
Alignment and variant calling of the DNA sequencing read followed the Genome Analysis Toolkit (GATK) best practice guidelines (https://software.broadinstitute.org/gatk/best-practices/)[47]. Default parameters were used for BWA-mem (http://bio-bwa.sourceforge.net/) and Picard Tools MarkDuplicates (https://broadinstitute.github.io/picard/Variant) to align reads and to mark PCR duplicates, respectively. GATK was used to realign indels, recalibrate quality scores, and generate GVCF files for each sample using the HaplotypeCaller tool. To minimize the potential downstream effects of differential coverage between the different capture platforms, a target bed file was created using the intersection of target regions of the three capture platforms (IDT xGen V1, Agilent SureSelect All Exon V7, and SeqCap EzExome V2). Case and control samples were called jointly using GATK GenotypeGVCF tools, and variant score recalibration was applied to all called variants. Passing variants were then annotated using the RefSeq hg19 gene definitions and databases using ANNOVAR[48].

### Quality control of de novo variants
Parent–child trios were excluded if unexpected family relationships were identified using relatedness statistics[49]. Trios were also omitted if the children were observed to have an outlier number of de novo variants (>20). PLINK/SEQ istats (https://zzz.bwh.harvard.edu/plinkseq/) was used to generate quality control statistics for both cases and controls, and principal component analyses were used to remove outliers (see Supplementary Fig. 1 and Supplementary Data 1 for details). After these quality control steps, we analyzed 147 parent–child trios with ADHD and 780 unaffected parent–child trios for de novo variants. The probands with ADHD included 25 females and 122 males, and the controls included 431 females and 349 males.

We then used stringent thresholds to assess de novo variants[44]. Specifically, we identified de novo variants as those that were heterozygous in the child (with an alternate allele frequency between 0.3 and 0.7) and not present in both parents (with an alternate allele frequency < 0.05). For variants on the X chromosome, de novo variants in male children were absent in the mother; de novo variants in female children were absent in both parents. For all de novo variants, we also required a sequencing depth of ≥20 in all family members at the variant position, alternate allele depth ≥5, and mapping quality ≥30. Calls were limited to one variant per person per gene, retaining variants with the most severe consequence[20]. We filtered to include rare de novo variants with an allele frequency <0.001 (0.1%) in the "non-neuro" subset of the Genome Aggregation Database (gnomAD v2.2.1). Within this set of rare de novo variants, we defined an ultra-rare subset as having an allele frequency of <0.00005 in the non-neuro subset of gnomAD[50]. The gnomAD v2.2.1 non-neuro dataset contains exome sequencing data from 104,068 individuals who were not ascertained for having a neurologic or psychiatric condition in case–control studies. All rare de novo damaging variants entering into our analyses were confirmed as present in the proband and absent in the parents by visualizing aligned sequencing reads from binary alignment map (.bam) files using the Integrative Genomics Viewer (IGV, https://igv.org/)[51,52] (see Supplementary Methods, Supplementary Figs. 2 and 3).

## Mutation rate analysis

To minimize potential bias in variant calling that may occur between different exome capture platforms and sequencing batches, especially between cases and controls, our primary comparisons are between mutation rates (per bp) within the "callable" exome per family. To calculate the callable exome denominator for our rates, we first used the GATK DepthofCoverage tool to count the number of bp in each trio that met the following criteria: sequenced at ≥20×, base quality ≥20, and mapping quality ≥30; these thresholds are the minimum required for de novo variant calling, described above. Additionally, a "callable" bp must be located within the intersection of the capture platforms (target intervals bed files) used for whole-exome DNA sequencing of ADHD cases and controls. The number of callable bases per family is listed in Supplementary Data 1 (1.1, "countable_coverage"). For each mutation class (e.g., synonymous, missense, PTV), the number of mutations was divided by the sum of callable bp for all trios; this rate was divided by 2 to calculate the haploid mutation rate for each mutation class. This was calculated separately for cases and controls. Confidence intervals were calculated (pois.conf.int, pois.exact functions from epitools v0.5.10.1 in R), and we used a one-tailed rate ratio test to compare de novo mutation rates between cases and controls (rateratio.test v1.1 in R).

Based on studies of other childhood-onset neuropsychiatric conditions[11,18–20], we hypothesized that mutation rates for rare and ultra-rare de novo protein-truncating variants (PTV) and damaging missense variants (Mis-D) would be greater for cases compared to controls. Mis-D variants were identified using the integrated "missense badness, PolyPhen-2, constraint" (MPC)[21] score > 2 as done in other recent studies[8,19,53]. The combined group of de novo PTV and Mis-D variants were considered "damaging" variants.

## Transmission and de novo association test analysis

We used a Bayesian extension of the original transmission and de novo association test (extTADA)[22] to integrate de novo and case–control variants in a hierarchical model to increase the power of identification of risk genes for ADHD. We obtained mutation counts for PTVs and Mis-D variants (MPC > 2) from an independent case–control study including 3206 individuals with ADHD and 5002 typically developing controls who did not have diagnoses of autism or intellectual disability[8]. We ran extTADA following the code outlined at https://github.com/hoangtn/extTADA/blob/master/examples/extTADA_OneStep.ipynb[22]. ExtTADA uses a Markov chain Monte Carlo approach to calculate all parameters used as input in the traditional TADA[54] through sampling from the posterior in one step with resulting credible intervals. Parameter estimation led to the following estimates of (1) proportion of risk genes ($\pi$) (lower-upper credible intervals): 5.50% (1.24–14.27%); (2) average relative risk ($\gamma$) (lower-upper credible intervals): Mis-D de novo = 20.34 (1.05–66.16), PTV de novo = 21.35 (3.08–66.56), Mis-D case–control = 1.61 (1.00–4.64), PTV case–control = 1.78 (1.08–4.91); and (3) variability in relative risk estimates per gene ($\beta$) (lower-upper credible intervals): Mis-D de novo = 0.83, PTV de novo = 0.82, Mis-D case–control = 6.17, PTV case–control = 3.98. These parameters were used by the extTADA function to calculate the Bayes factor and q-values (false discovery rate, FDR) for each gene (Supplementary Data 3). We applied commonly used statistical thresholds to define "potential" (FDR < 0.3) and "high confidence" (FDR < 0.1) risk genes[18].

To calculate the absolute number of ADHD risk genes, we multiplied the total number of genes included in the extTADA analysis (19,560) by the proportion of risk genes estimated by the extTADA pipeline. All genes from the list generated by denovolyzeR[55] except for American College of Medical Genetics genes (ACTA2, ACTC1, APC, APOB, ATP7B, BMPR1A, BRCA1, BRCA2, CACNA1S, COL3A1, DSC2, DSG2, DSP, FBN1, GLA, KCNH2, KCNQ1, LDLR, LMNA, MEN1, MLH1, MSH2, MSH6, MUTYH, MYBPC3, MYH11, MYH7, MYL2, MYL3, NF2, OTC, PCSK9, PKP2, PMS2, PRKAG2, PTEN, RB1, RET, RYR1, RYR2, SCN5A, SDHAF2, SDHB, SDHC, SDHD, SMAD3, SMAD4, STK11, TGFBR1, TGFBR2, TMEM43, TNNI3, TNNT2, TP53, TPM1, TSC1, TSC2, VHL) were included in the exTADA analysis.

## Gene set overlap

We examined if our list of genes with ultra-rare de novo damaging variants (PTV or Mis-D) in the ADHD probands overlapped with genes implicated in other DNA sequencing studies and genome-wide association studies. The Gene4Denovo database[23] (http://www.genemed.tech/gene4denovo/home) integrates de novo mutations from 68,404 individuals across 37 different phenotypes, including several neuropsychiatric conditions, but not including ADHD. We assessed the overlap between the Gene4Denovo gene list (release version updated 07/08/2022) and our list of genes with ultra-rare damaging de novo variants. The GWAS Catalog[56,57] was used to examine if this same list of genes harboring de novo damaging variants overlapped with loci mapped to genes in previous genome-wide association studies of neuropsychiatric phenotypes. The GWAS Catalog identifies past studies through weekly PubMed searches and extracts data for SNPs with $p < 1 \times 10^{-5}$ in the overall (initial GWAS + replication) population. All curated trait descriptions in the GWAS Catalog are mapped to terms from the experimental factor ontology (EFO), which provides a systematic description of traits to support the annotation, analysis, and visualization of data. We limited our overlap analysis to traits in the GWAS Catalog categorized under the umbrella terms 'nervous system disease' or 'psychiatric disorder' (additional details found at https://www.ebi.ac.uk/gwas//docs).

## Exploratory pathway analysis

We used ConsensusPathDB[58] (http://cpdb.molgen.mpg.de/, Latest Release 35, 05.06.2021) to assess whether our list of 23 genes with ultra-rare de novo damaging variants in ADHD probands (Table 1) was over-represented in gene-ontology and biological pathway sets. This network tool integrates information from 31 public databases. The following default settings were used for the exploratory gene set over-representation analysis: pathways as defined by pathway databases; select all databases; minimum overlap with input list = 2; p-value cut-off = 0.01; Gene ontology categories levels 2–5; select all biological process, molecular function, and cellular component categories; p-value cutoff = 0.01. P-values within each database are calculated using Fisher's exact test, corrected for multiple comparisons. In addition, ConsensusPathDB calculates q-values that are corrected for the number of tests performed across all databases. Q-values are computed as Benjamini–Hochberg (BH)-corrected values from the p-values of Fisher's exact tests, using the formula $q = (p*N)/r$, where N is the total number of tests performed (i.e., number of gene-ontology categories or pathways tested across all databases), and r is the rank of the p-value in the sorted list of all p-values. Supplementary Data 6 provides p-values < 0.01 and q-values for all enriched gene-ontology and pathway sets.

## Reporting summary

Further information on research design is available in the Nature Portfolio Reporting Summary linked to this article.

# Data availability

The DNA variant data generated in this study for parent–child trios with ADHD has been deposited in the NIH Database of Genotypes and Phenotypes (dbGaP) under accession number phs003647.v1.p1 and is available at the following URL. The raw DNA sequencing data are not openly available due to sensitivity reasons but are available from the corresponding author upon reasonable request. Responses to requests can be expected within one month and may require IRB approval and a data use agreement. Processed DNA sequencing results

are available in the supplementary information of the manuscript (Supplementary Data 1 and Supplementary Data 2). Control trio DNA sequencing data was obtained from the NIMH Data Archive (https://nda.nih.gov/edit_collection.html?id=2042). Several publicly available databases and datasets were used in the analyses, including the Genome Aggregation Database (gnomAD v2.2.1), Gene4Denovo (http://genemed.tech/gene4denovo/download, release version 07/08/2022), the GWAS Catalog (https://www.ebi.ac.uk/gwas/, accessed 03/24/2023), and ConsensusPathDB (release version 35, http://cpdb.molgen.mpg.de/). Source data are provided in this paper.

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

## Acknowledgements

We thank all of the families who contributed to this research study. We would also like to thank Kyle Satterstrom, Anders Børglum, and Mark Daly for sharing their case–control PTV and Mis-D counts for this analysis and all of the team members who contributed to data collection, including Chelsea Dale and Juliana Acosta. This work was supported by a Klingenstein Third Generation Foundation ADHD Fellowship grant (E.O.), a Yale Child Study Center Faculty Development Award (T.V.F.), and the Allison Family Foundation (T.V.F.). Brazilian samples were recruited and collected with support from the São Paulo Research Foundation (FAPESP), grant 2016/22455-8 (G.V.P.). E.O. was supported by the National Institute of Health (NIH) grants R25MH077823 (P.I. Martin), T32MH018268 (P.I. Crowley), and K08MH128665 (E.O.). L.C.F. was supported by São Paulo Research Foundation (FAPESP) grant #2021/08540-0 (L.C.F.). J.P. was supported by the Bradley Hospital COBRE Center for Sleep and Circadian Rhythms in Child and Adolescent Mental Health (P20GM139743, PI Carskadon). C.C. was supported by NIH grant K99MH128540(C.C.). J.L.K. and G.Z. were supported by the Larry and Judy Tanenbaum Family Foundation. Seventy-two of the sequenced ADHD parent–child trios were from the Génome Québec Genizon Biobank. The content is solely the responsibility of the authors and does not necessarily represent the official views of the NIH. Simons Simplex Collection (SSC) control parent–child trio genetic data used in the preparation of this manuscript were obtained from the NIH-supported National Database for Autism Research (NDAR). NDAR is a collaborative informatics system created by the National Institutes of Health to provide a national resource to support and accelerate research in autism. Dataset identifier: https://nda.nih.gov/edit_collection.html?id=2042. This manuscript reflects the views of the authors and may not reflect the opinions or views of the NIH or of the Submitters submitting original data to NDAR. We are grateful to all of the families at the participating SSC sites, as well as the principal investigators (A. Beaudet, R. Bernier, J. Constantino, E. Cook, E. Fombonne, D. Geschwind, R. Goin-Kochel, E. Hanson, D. Grice, A. Klin, D. Ledbetter, C. Lord, C. Martin, D. Martin, R. Maxim, J. Miles, O. Ousley, K. Pelphrey, B. Peterson, J. Piggot, C. Saulnier, M. State, W. Stone, J. Sutcliffe, C. Walsh, Z. Warren, E. Wijsman). We appreciate obtaining access to phenotypic data on SFARI Base. Approved researchers can obtain the SSC population dataset described in this study (https://www.sfari.org/resource/simons-simplex-collection/) by applying at https://base.sfari.org.

## Author contributions

E.O., J.P., G.V.P., C.C., J.L.K., and T.V.F. designed the research. E.O., L.C.F., W.L., L.A.V., G.Z., M.O.L., J.P., G.V.P., C.C., J.L.K., and T.V.F. performed the research. E.O., L.C.F., and T.V.F. analyzed the data and wrote the paper. All authors critically reviewed the paper.

## Competing interests

In the past 3 years, G.V.P. has been a consultant, advisory board member, and/or speaker for Aché, Abbott, Apsen, Medice, Novo Nordisk, Pfizer, and Takeda. G.V.P. also receives royalties from Editora Manole. J.L.K. is a member of the Scientific Advisory Board of Myriad Neuroscience, and author on several patents that are unrelated to the subject matter of this paper. The remaining authors declare no competing interests.
