## [Peer Review File · Nature Communications]

Rare de novo damaging DNA variants are enriched in attention-deficit/hyperactivity disorder and implicate risk genesReviewers' Comments:

Reviewer #1:

Remarks to the Author:

Many studies leverage Next-generation sequencing data to identify risk alleles and rare mutations in complex disorders. A common approach involves the analysis of familial Whole Exome or Whole Genome sequencing data, applying different inheritance analysis modes, including identifying rare de-novo variants (DNV). The flagged genes or variants are then subjected to additional analysis, including genotype/phenotype correlation, comparison to controls and other datasets in addition to Gene Ontology and pathway enrichment.

Using this approach, the authors analysed 147 ADHD Trios (Simplex Cases) for ultra-rare de novo variants in comparison to 780 controls and identified 25 ultra-rare de novo variants in 24 genes. Although the overall rate of de-novo variants was similar in both groups, there was an enrichment in ultra-rare de-novo damaging variants in ADHD trios compared to controls.

Transmission and De novo Association test (extTADA) using a set of (>3000 ADHD and >5000 controls) identified 862 risk genes and flagged a few genes of interest. Among these genes, KDM5B was identified as a significantly associated gene with two ultra-rare damaging variants in this cohort. As expected, the identified genes overlapped with other neurodevelopmental disorders (NDD). The analysis in this study focused on identifying genetic variants without correlation with clinical phenotype or sex/ gender.

The information presented in this manuscript adds to the growing genetic data in the field. The authors presented their data clearly, and all supplementary figures and tables were informative. However, there are a few points to consider.

1- the authors present the QC metrics for WES data; however, there are frameshift variants in the flagged 25 ultra-rare de novo variants. were these variants confirmed by Sanger sequencing?

2- it would be interesting to know if there were any overlapping genes when looking at the overall DNV in both ADHD Trios and controls.

3- Page7/lines 210-212 (CTNND2 and EML6, had a de novo PTV variant in one individual with ADHD and a de novo missense variant that was not predicted to be damaging according to an MPC<2, but the CTNND2 variant was predicted to be damaging by PolyPhen2) The statement is confusing. If the authors can rephrase.

4- Lastly, the authors should add a column to Table 1 (main) indicating if the gene was previously reported in ADHD or in another NDD.

Reviewer #2:

Remarks to the Author:

The study by Olfson et al., demonstrates for the first-time enrichment of ultra-rare de novo deleterious variants in individuals with ADHD compared to controls in analyses of 152 trios where the proband has ADHD compared to 788 control parent-child trios without ADHD. Additionally, the authors combine the ADHD-trio identified de novo variants with counts from published exome sequencing data (from Satterstrom et al. Nature Neuroscience 2019) on 3,206 ADHD cases and 5,002 controls and identify KDM5B as a potential risk gene for ADHD (FDR = 0.05) and suggest two other risk genes POMT1 (FDR=0.21), and YLPM1 (FDR=0.28).

In the trio analysis, the authors identify 24 genes with de novo mutations and use these genes in pathway analyses and explore their relation to GWAS findings and findings from other sequencing studies.

The study is methodologically well done, the manuscript is clear, and the result showing increased

deleterious de novo variants in ADHD compared controls is a new and interesting finding. However, besides from this message, the gene-based findings are weak except from the KDM5B finding, which means that several of the other results should be taken with caution. The authors are aware of this and states this clearly in the manuscript.

I have some additional points below and please note that when I write "deleterious variants" I refer to PTVs+D-miss variants.

Page 5. The authors write "25 genes with ultra-rare de novo damaging variants" and refer to Table 1. However, in Table 1 only 24 genes are listed – please clear this.

Please give information in the main text about the number of deleterious rare/ultra-rare de novo variants identified in the control trios.

It has previously been demonstrated that PTVs comprise a larger part of the total number of de novo variants than when considering transmitted variants (e.g., Figure 1.a, Satterstrom et al. cell 2020). Do you observe the same in the ADHD trios analyzed in this study?

On Figure 1 the authors give mutation rates on the y-axis. Are those rates in the range of what we would expect from other studies of the mutation rate in the human genome?

It is not clear to me if variants on the sex-chromosomes were included. Sex-chromosome variants were not analyzed in the previous ADHD exome-sequencing study by Satterstrom et al. and I assume sex chromosome variants are not included here?

Most of the supplementary tables lack description of the column headers: "GWAS catalog overlap", "gene4denovo", "Tada results" and "pathway results". Please add that. And the "sample info" table only has "key" for one sheet. Please add information about the other sheets.

Also please add Supplementary Table number in the excel sheets, that would help the reader.

One page 9 the authors write "We prioritized the study of simplex (no known family history of ADHD) 280 parent-child trios to increase the likelihood of detecting de novo variants." Please correct to "... the likelihood of detecting deleterious de novo variants". Or use another word than "deleterious" if better.

Please add information about metrics describing how evolutionarily constrained the genes identified in the de novo analysis are (e.g., pLI or Loeuf score). I think this information is important to evaluate the results. Especially since several of the 24 genes might not be ADHD risk genes and thus the pLI score could help highlight genes of more interest than others.

It could be considered to cut some of the discussion regarding specific genes with de novo deleterious variants considering the low statistical support for their involvement in ADHD (or only include discussion of those that are evolutionarily constrained).

Pathway exploratory analyses – please provide information about how you corrected for multiple testing within databases and across databases. Additionally, in the text related to Supplementary Table 6 (the pathway analyses) is written that "all sets with a p-value <0.01 are listed". Is that p-value corrected for multiple testing? Also please provide information in the text related to the table about how many genes were included in the analysis (I assume the 24 genes listed in Table 1).

Reviewer #3:

Remarks to the Author:

The authors find significant enrichment of de novo variants (PTVs, likely damaging missense variants) in ADHD, following the path laid out by various other neurodevelopment disorders. Overall, the findings align with expectation, and the authors have clearly noted the limitations imposed by the current sample size, which renders in-depth analyses in some areas not practical.

I only have a few comments as follows:

Major

1. Could we request more details on the calculation of mutation rates? It is currently slightly unclear in the methods section. Have these estimate mutation rates been compared to gnomAD reported mutation rates? Given the small sample sizes and observed variant counts, the mutation rate is unlikely to have significant impact on the FDRs produced by TADA - however, as sample sizes increases, being accurate with the mutation rate is very important for the FDRs to be calibrated.

Minor

1. Unaffected SSC siblings are likely still elevated for genetic variant load versus a population cohort - however, the SSC siblings are commonly used across a myriad of phenotypes as controls, so generates an advantage during cross-disorder comparisons.

Suggestions:

1. What is the ADHD patient sex breakdown? Understanding that ADHD is diagnosed more often in boys vs girls and in men vs women, as well as previously publications that there is detectible common genetic variants to explain this discrepancy, would this study be powered to examine this question?
2. Are there plans to delineate CNVs, even if just rare, from these individuals?
3. What inference about the genetic architecture differences between ADHD and other NDDs can be drawn? In particular, are the PTV signals enriched in constrained genes (measured by LOEUF or pLI) versus unconstrained genes?
4. With the recent publication of alpha missense, we have observed some orthogonal value in incorporating it as a variant prioritization measure in addition to MPC - perhaps this could help elevate additional missense driven signals.

Response to Reviewers

Thank you for your interest in a revised version of our manuscript entitled “Rare *de novo* damaging DNA variants are enriched in attention-deficit/hyperactivity disorder and implicate risk genes.” We appreciate the positive comments and helpful suggestions from the three reviewers, and we address each of their concerns in detail below.

Reviewer 1:

Thank you for commenting that “*this manuscript adds to the growing genetic data in the field*” and “*the authors presented their data clearly, and all supplementary figures and tables were informative.*”

Comment 1.1: “*the authors present the QC metrics for WES data; however, there are frameshift variants in the flagged 25 ultra-rare denovo variants. were these variants confirmed by Sanger sequencing?*”

Response 1.1: This is an important point, and we apologize for omitting our confirmation methods from the original manuscript. In this study, we applied stringent variant calling thresholds for identifying *de novo* mutations (explained in Methods) and we now add data from our previous studies validating this approach by demonstrating a low false positive call rate by Sanger sequencing. Additionally, in our revised manuscript, we now present *in silico* confirmation data for all rare *de novo* variants, showing aligned sequencing reads (.bam files) of parent-child trios using the Integrative Genomics Viewer. We now include these alignment plots for all rare and ultra-rare damaging *de novo* variants in the Supplementary Information (Supplementary Figures 2-3).

Below is an example of one plot from Supplementary Figure 2, showing the presence of a frameshift deletion (chr11:65359271 CATGG>C, highlighted by the red bar) in the proband (top panel) and absence in both parents:

As shown in Supplementary Figures 2-3, visual inspection of 24 rare variants in ADHD trios is consistent with true *de novo* variants, and two variants (one rare and one ultra-rare frameshift indel) did not pass confirmation. We now include our confirmation strategy by adding the following text to the Methods section:

“All rare *de novo* damaging variants entering into our analyses were confirmed as present in the proband and absent in the parents by visualizing aligned sequencing reads from binary alignment map (.bam) files using the Integrative Genomics Viewer (IGV, <https://igv.org/>) (see Supplementary Methods, Supplementary Figures 2-3).”

We also added data in support of this strategy in the Supplementary Information (Supplementary Methods):

“We confirmed all rare *de novo* damaging variants entering into our analyses (Table 1, Supplementary Data 2) by visualizing aligned sequencing reads using the Integrative Genomics Viewer (IGV, <https://igv.org/>)² for the proband, father, and mother (Supplementary Figure 2). *In silico* visualization increases confidence in variant calls and reduces the risk of false positives¹. Aggregate data from our earlier studies using the

same analytical pipeline and the same stringent thresholds for calling *de novo* SNVs and indels support this approach^{3,4}. In these studies, we confirmed 192/193 (99.5%) *de novo* SNVs and 7/8 (87.5%) *de novo* indels by Sanger sequencing, finding that *in silico* visualization was consistent with Sanger confirmation status. Although DNA from Simons Simplex Collection control subjects and from Genizon biobank ADHD samples were not available for confirmation by Sanger sequencing, prior Sanger validation of our *in silico* confirmation methods and our application of joint calling with stringent calling thresholds in cases and controls make it unlikely that our findings are biased by false positive calls.”

Comment 1.2: “*it would be interesting to know if there were any overlapping genes when looking at the overall DNV in both ADHD Trios and controls.*”

Response 1.2: For genes harboring all rare *de novo* variants, there are several overlapping genes between ADHD and control trios. This information is now provided in Supplementary Data 2 (2.3). Of the rare *de novo* predicted-damaging (PTV or Mis-D) variants in ADHD cases, two of these genes (*CNTNND2*, *USP54*) also harbored rare *de novo* missense variants in controls. However, these variants in controls were not predicted to be damaging by MPC score. We have added text to the Results section to highlight that we found no overlap between cases and controls among genes harboring predicted-damaging rare or ultra-rare *de novo* variants.

“The list of genes harboring rare or ultra-rare *de novo* damaging variants in ADHD cases did not overlap with genes harboring rare or ultra-rare *de novo* damaging variants in control parent-child trios passing quality control (**Supplementary Data 2**).”

Comment 1.3: “*Page 7/lines 210-212 (CNTNND2 and EML6, had a de novo PTV variant in one individual with ADHD and a de novo missense variant that was not predicted to be damaging according to an MPC<2, but the CNTNND2 variant was predicted to be damaging by PolyPhen2) The statement is confusing. If the authors can rephrase.*”

Response 1.3: We apologize for the confusion about this statement. We rephrased this statement to convey that we found two additional genes (*CNTNND2* and *EML6*), each harboring ultra-rare *de novo* variants in two ADHD cases. For each gene, we found a *de novo* PTV in one individual and a *de novo* missense variant in a second individual with ADHD. Both *de novo* missense variants were not predicted to be damaging, according to the MPC >2 threshold applied in this study. However, these *de novo* missense variants were predicted to be possibly damaging (*EML6*) or probably damaging (*CNTNND2*) according to a different (less stringent) metric commonly used in WES studies, PolyPhen2-HDIV. These metrics are included in Supplementary Data 2 (2.1). Please note that this statement has been moved to the Supplementary Discussion in response to a suggestion by Reviewer 2 (Comment 2.10). To clarify the statement, we revised the text in the Supplementary Information (Supplementary Discussion) as follows:

“*CNTNND2* (FDR=0.26) and *EML6* each had a *de novo* PTV variant in one individual with ADHD and a *de novo* missense variant in a second individual that was not predicted to be damaging according to MPC<2. However, it is interesting to note that these *de novo* missense variants were predicted to be possibly damaging (*EML6*) or probably damaging (*CNTNND2*) using a different (less stringent) metric commonly used in WES studies, PolyPhen2-HDIV (**Supplementary Data 2**).”

Comment 1.4: “*Lastly, the authors should add a column to Table 1 (main) indicating if the gene*

was previously reported in ADHD or in another NDD.”

Response 1.4: Thank you for this suggestion. We agree that this is useful information, and we have added a column to Table 1 showing if the gene was previously reported as a likely risk gene (FDR<0.05) for NDDs in the largest and most recent meta-analysis of autism spectrum disorder and developmental delay (Fu et al., *Nature Genetics*, 2022). To our knowledge, no previous studies have specifically identified high-confidence risk genes for ADHD. Additionally, we added the following text to the Results section to highlight this information:

“Six of these 23 genes were recently reported as likely risk genes (FDR<0.05) for neurodevelopmental disorders (NDD) in the largest meta-analysis of ASD and developmental delay⁹, including *FBXO11* (FDR=0), *KDM5B* (FDR=0), *STAG1* (FDR=1.98x10⁻⁷), *CTNNA2* (FDR=9.49x10⁻⁵), *EML6* (FDR=0.002), and *PAK1* (FDR=0.006) (**Table 1, Supplementary Data 4**).”

Reviewer 2:

We thank this reviewer for noting, “*The study is methodologically well done, the manuscript is clear, and the result showing increased deleterious de novo variants in ADHD compared to controls is a new and interesting finding.*”

Comment 2.1: “*Page 5. The authors write “25 genes with ultra-rare de novo damaging variants” and refer to Table 1. However, in Table 1 only 24 genes are listed – please clear this.*”

Response 2.1: Thank you for bringing this to our attention. The reviewer is correct; we originally identified 25 ultra-rare *de novo* damaging variants in 24 unique genes, with *KDM5B* appearing twice. In our revised manuscript we now base all analyses on validated *de novo* variants (see Response 1.1), so this number has now changed to 23 genes. We corrected this sentence and another occurrence in the Results section of the manuscript to reflect 23 genes with ultra-rare *de novo* damaging variants.

“Using the list of 23 genes with ultra-rare *de novo* damaging variants (PTV and Mis-D) in 147 ADHD probands (**Table 1, Supplementary Data 2**), we identified overlap with risk genes for other conditions.”

“Using this same list of 23 genes harboring ultra-rare *de novo* damaging variants in ADHD trios, we also conducted exploratory analyses to identify enriched gene ontology and biological pathways. Several gene ontology and pathway-based sets were enriched for these 23 genes identified in ADHD (**Supplementary Data 6**).”

Comment 2.2: “*Please give information in the main text about the number of deleterious rare/ultra-rare de novo variants identified in the control trios.*”

Response 2.2: We revised text in the Results section to include counts of deleterious (damaging) *de novo* variants in the control trios.

“Among 147 ADHD parent-child trios passing quality control, we identified 24 ultra-rare *de novo* damaging variants in 23 individuals (**Table 1, Supplementary Data 2**). Among 780 control parent-child trios passing quality control, we identified 51 ultra-rare *de novo* damaging variants in 50 individuals (**Supplementary Data 2**).”

Also, for readers who wish to view detailed variant annotation information, we now include additional columns in the Supplementary Data 2 tables (e.g., variant_type, damaging, mis_mpc_type, ultra-rare) that enable easier filtering to display damaging (PTV, Mis-D) and ultra-rare (gnomAD non-neuro allele frequency <0.00005) *de novo* variants in cases and controls.

Comment 2.3: “It has previously been demonstrated that PTVs comprise a larger part of the total number of *de novo* variants than when considering transmitted variants (e.g., Figure 1.a, Satterstrom et al. *cell* 2020). Do you observe the same in the ADHD trios analyzed in this study?”

Response 2.3: Thank you for bringing this interesting finding to our attention. Just as Satterstrom et al. (*Cell* 2020, Figure 1A, Table S1) show that rare PTVs comprise a larger proportion of all rare *de novo* variants (13.63%) compared to all rare transmitted variants (3.39%) in autism, our results show a similar pattern in ADHD. In our data, ultra-rare PTVs comprise 15.18% of all ultra-rare *de novo* variants compared to 7.64% of ultra-rare transmitted variants. We now mention this finding in the Discussion and include the counts and percentages as a new tab in Supplementary Data 2 (2.5):

“Similar to ASD (Satterstrom et al., 2020), we found that PTVs comprise a greater proportion of rare *de novo* variants than transmitted variants in ADHD (**Supplementary Data 2**).”

Comment 2.4: “On Figure 1 the authors give mutation rates on the y-axis. Are those rates in the range of what we would expect from other studies of the mutation rate in the human genome?”

Response 2.4: Mutation rates can differ among studies depending on the specific thresholds and filters applied to obtain the variant classes (e.g., Mis-D, Damaging) and frequencies being investigated. Compared with recent studies using similar methods and variant class definitions, we do find similar *de novo* mutation rates in our sample.

For example, our rate of rare *de novo* PTV variants is 0.21×10^{-8} (95% CI 0.13×10^{-8} - 0.33×10^{-8}) per base pair (bp) in 147 ADHD parent-child trios and 0.14×10^{-8} (95% CI 0.10×10^{-8} - 0.18×10^{-8}) per bp in 780 control trios (Figure 1, Supplementary Table 1). The definition of rare *de novo* PTV variants applied in our study aligns with the definition (LGD) applied in a whole-exome sequencing study of OCD (Cappi et al., *Biological Psychiatry*, 2020). This study reported a similar rare *de novo* PTV (LGD) mutation rate of 0.22×10^{-8} per bp in 184 OCD parent-child trios and 0.11×10^{-8} in 777 control trios using similar methods to calculate mutation rates. The other comparable class of variants is rare *de novo* synonymous variants. Our rates in cases and controls are 0.35×10^{-8} (0.24 - 0.50×10^{-8}) and 0.45×10^{-8} (0.38 - 0.52×10^{-8}) per base pair, respectively. This is in line with rare *de novo* synonymous rates from Cappi et al. (2020) of 0.49×10^{-8} (0.36 - 0.65×10^{-8}) and 0.50×10^{-8} (0.43 - 0.57×10^{-8}) per bp.

In addition, Wang et al. (*Cell Reports*, 2018) reported a rare *de novo* PTV (LGD) mutation rate of 0.17×10^{-8} ($\pm 0.047 \times 10^{-8}$) per bp in 777 Tourette syndrome parent-child trios and 0.093×10^{-8} ($\pm 0.029 \times 10^{-8}$) per bp in 1,153 control trios. Rare *de novo* synonymous mutation rates from this study were 0.42×10^{-8} ($\pm 0.075 \times 10^{-8}$) per bp in cases and 0.40×10^{-8} ($\pm 0.060 \times 10^{-8}$) per bp in controls.

We are unable to directly compare our rates of *de novo* damaging missense (Mis-D) variants with these two papers because we used MPC scores to identify Mis-D variants; this criterion is more stringent than the PolyPhen2-HDIV score criterion used in these other papers. However, a smaller study led by our group (Olfson et al. *Anxiety and Depression*, 2022) also

used MPC score >2 to identify rare *de novo* Mis-D variants, reporting a mutation rate of 0.12×10^{-8} (0.04-0.28 $\times 10^{-8}$) per bp in 68 anxiety parent-child trios and 0.05×10^{-8} (0.03-0.08 $\times 10^{-8}$) per bp in 783 controls. This is in line with our data in the current ADHD study: 0.08×10^{-8} (0.03-0.16 $\times 10^{-8}$) per bp in cases and 0.05×10^{-8} (0.03-0.07 $\times 10^{-8}$) per bp in controls.

To highlight the fact that our rates are in the range of what has been reported in other studies, we added the following to the Discussion:

“It is important to note that these estimated enrichments have wide confidence intervals, so caution is warranted in interpreting these results, and replication in larger ADHD parent-child trio cohorts is needed. Nevertheless, our observed mutation rates and our enrichment of rare *de novo* PTV and Mis-D variants and mutation rates are of a similar magnitude to those reported in other parent-offspring trio studies examining *de novo* variation in other neurodevelopmental disorders, including ASD and Tourette’s disorder.”

Comment 2.5: *“It is not clear to me if variants on the sex-chromosomes were included. Sex-chromosome variants were not analyzed in the previous ADHD exome-sequencing study by Satterstrom et al. and I assume sex chromosome variants are not included here?”*

Response 2.5: We apologize this was not clear in the original manuscript. We did include variants on the X chromosome, and the case-control PTV and Mis-D counts provided to us by Kyle Satterstrom, Anders Børglum, and Mark Daly also included ultra-rare variants on the X chromosome. Therefore, we were able to combine the Satterstrom et al. X chromosome data with our X chromosome *de novo* variant counts to conduct the extTADA analysis.

To clarify that our analyses included variants on the X chromosome, we added the following to the Methods:

“...we identified *de novo* variants as those that were heterozygous in the child (with an alternate allele frequency between 0.3 and 0.7) and not present in both parents (with an alternate allele frequency < 0.05). For variants on the X chromosome, *de novo* variants in male children were absent in the mother; *de novo* variants in female children were absent in both parents.”

Comment 2.6: *“Most of the supplementary tables lack description of the column headers: “GWAS catalog overlap”, “gene4denovo”, “Tada results” and “pathway results”. Please add that. And the “sample info” table only has “key” for one sheet. Please add information about the other sheets.”*

Response 2.6: Thank you for this suggestion. We now include “Key” tabs describing the column headings for all supplementary tables. This tab is always the first tab of the Supplementary Data files.

Comment 2.7: *“Also please add Supplementary Table number in the excel sheets, that would help the reader.”*

Response 2.7: We now include Supplementary Table numbers and titles at the top of each sheet. We also include the table numbers within the Excel sheet tab labels of the supplementary data files.

Comment 2.8: *“One page 9 the authors write “We prioritized the study of simplex (no known family history of ADHD) parent-child trios to increase the likelihood of detecting de novo*

variants.” Please correct to “... the likelihood of detecting deleterious *de novo* variants”. Or use another word than “deleterious” if better.”

Response 2.8: We have changed this sentence in the Methods section, as suggested:

“We prioritized the study of simplex (no known family history of ADHD) parent-child trios to increase the likelihood of detecting deleterious *de novo* variants.”

Comment 2.9: “Please add information about metrics describing how evolutionarily constrained the genes identified in the *de novo* analysis are (e.g., pLI or Loeuf score). I think this information is important to evaluate the results. Especially since several of the 24 genes might not be ADHD risk genes and thus the pLI score could help highlight genes of more interest than others.”

Response 2.9: Thank you for this suggestion. We have added pLI scores to Table 1 as suggested by the reviewer. In addition, we now provide pLI and LOEUF for each variant (when available) in Supplementary Data 2 (see tabs 2.1, 2.2). Please also see our response to Reviewer 3 (**Response 3.5**).

Comment 2.10: “It could be considered to cut some of the discussion regarding specific genes with *de novo* deleterious variants considering the low statistical support for their involvement in ADHD (or only include discussion of those that are evolutionarily constrained).”

Response 2.10: We agree with this suggestion and have now moved the discussion of several genes harboring *de novo* deleterious variants (including *YLPM1*, *CTNND2*, and *EML6*) to Supplementary Information (Supplementary Discussion). In the main Discussion, we now only discuss *KDM5B* and three genes (*FBXO11*, *STAG1*, and *CTNNA2*) with *de novo* deleterious variants, high constraint (pLI) scores, and previously identified as risk genes for neurodevelopmental disorders. These changes have been tracked in the main text and Supplementary Discussion.

Comment 2.11: “Pathway exploratory analyses – please provide information about how you corrected for multiple testing within databases and across databases. Additionally, in the text related to Supplementary Table 6 (the pathway analyses) is written that “all sets with a p-value <0.01 are listed”. Is that p-value corrected for multiple testing? Also please provide information in the text related to the table about how many genes were included in the analysis (I assume the 24 genes listed in Table 1).”

Response 2.11: Thank you for the opportunity to clarify the methods behind our exploratory pathway analyses. Within each database, Fisher’s exact test p-values are corrected for multiple comparisons, and we present all sets with $p < 0.01$ in Supplementary Data 6. In addition, ConsensusPathDB calculates q-values that incorporate the p-value and correct for the number of tests performed across all databases. Although we present all enrichment results with a corrected p-value of < 0.01 in Supplementary Data 6, we deliberately only mention pathways in the text where q-values are < 0.01 . We now detail these methods for our exploratory pathway analyses in the Methods section of the main text:

“We used ConsensusPathDB⁵⁶ (<http://cpdb.molgen.mpg.de/>, Latest Release 35, 05.06.2021) to assess whether our list of 23 genes with ultra-rare *de novo* damaging variants in ADHD probands (**Table 1**) was over-represented in gene-ontology and biological pathway sets. This network tool integrates information from 31 public databases. The following default settings were used for the exploratory gene set over-

representation analysis: pathways as defined by pathway databases; select all databases; minimum overlap with input list=2; p-value cutoff=0.01; Gene ontology categories levels 2, 3, 4, and 5; select all biological process, molecular function, and cellular component categories; p-value cutoff=0.01. P-values within each database are calculated using a Fisher's exact test, corrected for multiple comparisons. In addition, ConsensusPathDB calculates q-values that are corrected for the number of tests performed across all databases. Q-values are computed as Benjamini-Hochberg (BH)-corrected values from the p-values of Fisher's exact tests, using the formula $q=(p*N)/r$, where N is the total number of tests performed (i.e., number of gene-ontology categories or pathways tested across all databases), and r is the rank of the p-value in the sorted list of all p-values. **Supplementary Data 6** provides p-values < 0.01 and q-values for all enriched gene-ontology and pathway sets."

Reviewer 3:

Thank you for highlighting that our manuscript "*clearly noted the limitations imposed by the current sample size.*" We address the reviewer's specific comments below, including one "major," one "minor," and three "suggestions".

Comment 3.1: "*Major: Could we request more details on the calculation of mutation rates? It is currently slightly unclear in the methods section. Have these estimate mutation rates been compared to gnomAD reported mutation rates? Given the small sample sizes and observed variant counts, the mutation rate is unlikely to have significant impact on the FDRs produced by TADA - however, as sample sizes increases, being accurate with the mutation rate is very important for the FDRs to be calibrated.*"

Response 3.1: We apologize that the methods of our mutation rate calculations were unclear in the original manuscript. We have expanded the "Mutation rate analysis" section of the Methods to provide more details on our approach for calculating per base pair mutation rates:

"To minimize potential bias in variant calling that may occur between different exome capture platforms and sequencing batches, especially between cases and controls, our primary comparisons are between mutation rates (per bp) within the "callable" exome per family. To calculate the callable exome denominator for our rates, we first used the GATK DepthofCoverage tool to count the number of bp in each trio that met the following criteria: sequenced at $\geq 20x$, base quality ≥ 20 , and mapping quality ≥ 30 ; these thresholds are the minimum required for *de novo* variant calling, described above. Additionally, a "callable" bp must be located within the intersection of the capture platforms (target intervals bed files) used for whole-exome DNA sequencing of ADHD cases and controls. The number of callable bases per family is listed in Supplementary Data 1 (1.1, "countable_coverage"). For each mutation class (e.g., synonymous, missense, PTV), the number of mutations was divided by the sum of callable bp for all trios; this rate was divided by 2 to calculate the haploid mutation rate for each mutation class. This was calculated separately for cases and controls. Confidence intervals were calculated (pois.conf.int, pois.exact functions from epitools v0.5.10.1 in R), and we used a one-tailed rate ratio test to compare *de novo* mutation rates between cases and controls (rateratio.test v1.1 in R)."

We agree with Reviewer 3 that it is very important to be accurate with mutation rates, especially as sample sizes increase, to obtain properly calibrated FDRs in the TADA analysis. Because we

are calculating *de novo* mutation rates and applying stringent thresholds to call *de novo* variants (to minimize false positive calls), we are not able to use gnomAD to compare our mutation rates because gnomAD does not include parent-child trios needed for this comparison. However, we have compared our rates to those in studies that use a similar approach to variant calling and mutation rate calculations, and we are reassured by the similarities. Please see our response to Reviewer 2 (**Response 2.4**) for these *de novo* mutation rate comparisons.

For our TADA analysis, we combined our *de novo* variant counts with case-control variant counts from a larger independent ADHD sequencing study (described in Satterstrom et al., *Nature Neuroscience*, 2019). Notably, given their case-control design, Satterstrom et al. were able to compare their rates to those in gnomAD. As outlined in their Methods and in their Supplementary Table 5, Satterstrom et al. omitted variant counts in genes with greater synonymous mutation rates in cases compared to gnomAD. This effectively removed genes from the TADA analysis that may have been sequenced more deeply (and therefore more likely to detect mutations) in cases than in (gnomAD) controls.

Comment 3.2: *“Minor: Unaffected SSC siblings are likely still elevated for genetic variant load versus a population cohort - however, the SSC siblings are commonly used across a myriad of phenotypes as controls, so generates an advantage during cross-disorder comparisons.”*

Response 3.2: We agree with this statement and have added the following sentence to the limitations section of the Discussion:

“It is important to note that these SSC control siblings may have an elevated genetic variant load compared to a population cohort; mutation rates and gene enrichments reported in our ADHD cases would have to overcome this potential elevation in controls to achieve statistical significance. However, these SSC control siblings are often used in parent-child trio studies, offering an advantage for future cross-disorder comparisons.”

Comment 3.3: *“Suggestion: What is the ADHD patient sex breakdown? Understanding that ADHD is diagnosed more often in boys vs girls and in men vs women, as well as previously publications that there is detectible common genetic variants to explain this discrepancy, would this study be powered to examine this question?”*

Response 3.3: We apologize for omitting this data from our original submission. We have now added biological sex of each subject to Supplementary Data 1 (tab 1.1, Column C) and Supplementary Data 2 (tabs 2.1 and 2.2, Column B). After QC, our ADHD sample includes 25 females and 122 males; our control sample includes 431 females and 349 males. Given the small number of affected females in our sample, our statistical power to detect a sex-stratified differential mutation rate is low. Nevertheless, we now include these exploratory calculations as Supplementary Data 2 (tab 2.4). As expected, we did not detect a significant mutation rate difference in any variant type when comparing ADHD females to ADHD males or when comparing ADHD females to control females. Comparisons between ADHD males and control males continue to show increased mutation rates of *de novo* PTVs and damaging variants.

Comment 3.4: *“Suggestion: Are there plans to delineate CNVs, even if just rare, from these individuals?”*

Response 3.4: In the current study, we chose to focus on *de novo* SNVs and indels in our sample because there was a direct path to leverage our trio data for gene discovery by combining our data with the large SNV dataset from Satterstrom et al. (2019). We do plan to call

CNVs from WES data in our ADHD trios, but conducting this analysis properly with sufficient statistical power will require a larger sample size with sequencing read data (fastq or bam). This data format is not available from the Satterstrom et al. study. However, we are actively pursuing opportunities for accessing larger datasets which would enable a proper CNV analysis in ADHD.

Comment 3.5: *“Suggestion: What inference about the genetic architecture differences between ADHD and other NDDs can be drawn? In particular, are the PTV signals enriched in constrained genes (measured by LOEUF or pLI) versus unconstrained genes?”*

Response 3.5: We now provide comparisons between cases and controls while classifying PTVs by constraint metrics (pLI and LOEUF) in Supplementary Data 2. We added these metrics as annotation columns to the lists of rare *de novo* variants in cases (tab 2.1) and controls (tab 2.2). Mutation counts, rates (per base pair), rate ratios, and p-values are now presented in tab 2.6.

We observe an interesting trend toward increasing rate ratios when moving from relatively intolerant to tolerant PTVs in ADHD compared to controls. Specifically, ultra-rare *de novo* variants with pLI scores <0.5 (RR 2.39, $p=0.010$) and with LOEUF scores >0.35 (RR 1.92, $p=0.031$) are seen more frequently in our ADHD trios versus controls. This suggests a potential role for more tolerant/less constrained *de novo* PTVs in ADHD compared to other neurodevelopmental disorders. However, we are mindful of the limitations of our sample size when examining constrained *de novo* variants and the need for larger studies in ADHD parent-child trios to adequately assess this. Our hope is that the findings from this paper will motivate larger parent-child trio studies in ADHD in a similar manner to large-scale efforts in ASD and DD, which have significantly improved our understanding of the genomic architecture and risk genes in these other neurodevelopmental disorders.

Comment 3.6: *“Suggestion: With the recent publication of alpha missense, we have observed some orthogonal value in incorporating it as a variant prioritization measure in addition to MPC - perhaps this could help elevate additional missense driven signals.”*

Response 3.6: In response to this suggestion, we now include AlphaMissense scores and categories (pathogenic, benign, ambiguous) for rare *de novo* variants in Supplementary Data 2 (tabs 2.1 and 2.2). It is important to note that variant-level data required for AlphaMissense scores were not available from the Satterstrom et al. dataset, so we were unable to use these scores in our extTADA analysis. However, we now show variant counts, mutation rates, and rate ratios for the three AlphaMissense categories as an exploratory analysis in Supplementary Data 2 (tab 2.6). In general, AlphaMissense calls ~5x more missense variants as “pathogenic” compared to MPC score in our data. Similar to MPC, we find greater rate ratios for “pathogenic” missense rare and ultra-rare *de novo* variants in ADHD compared to controls, although this enrichment is not statistically significant on its own, similar to our MPC-based analysis. We also note that all rare *de novo* damaging missense variants (based on $MPC>2$) found in our ADHD cases were also scored as “pathogenic” by AlphaMissense. We appreciate the reviewer bringing these alternate scores to our attention, and we will continue to examine their utility in future analyses.

Reviewers' Comments:

Reviewer #1:

Remarks to the Author:

The authors thoroughly addressed the comments and incorporated the requested changes in a detailed manner. I have no additional comments to add.

Reviewer #2:

Remarks to the Author:

The authors have responded well to my comments and the new information and clarifications have added value to the manuscript and increased the quality.

I have one last minor comment. In studies of other psychiatric disorder (e.g. schizophrenia), we start to see the convergence in loci hit by both rare and common variants. This study find overlap with some genes implicated in other neurodevelopmental and psychiatric disorders but not common variant ADHD risk loci. This lack of overlap could be due to low samples size in the current study and that only 27 genome-wide significant loci have been identified for ADHD, but have the authors investigated enrichment of their 23 de novo risk genes among top-associated common variant risk genes (e.g. top 100 MAGMA genes), to evaluate more in-depth potential overlap with genes implicated in ADHD by common variants?

Reviewer #3:

Remarks to the Author:

The authors have thoughtfully responded to the queries put forth by the reviewers. Thank you.

Response to Reviewers

Thank you for your continued interest in a revised version of our manuscript, “Rare de novo damaging DNA variants are enriched in attention-deficit/hyperactivity disorder and implicate risk genes.” We also thank the reviewers for their positive comments.

There was one additional minor comment mentioned by **Reviewer 2**:

“I have one last minor comment. In studies of other psychiatric disorder (e.g. schizophrenia), we start to see the convergence in loci hit by both rare and common variants. This study find overlap with some genes implicated in other neurodevelopmental and psychiatric disorders but not common variant ADHD risk loci. This lack of overlap could be due to low samples size in the current study and that only 27 genome-wide significant loci have been identified for ADHD, but have the authors investigated enrichment of their 23 de novo risk genes among top-associated common variant risk genes (e.g. top 100 MAGMA genes), to evaluate more in-depth potential overlap with genes implicated in ADHD by common variants?”

Response: Thank you for this suggestion. The 23 genes harboring de novo gene-damaging mutations do not overlap with either the 76 prioritized risk genes identified by positional and functional annotation or the 45 exome-wide significant genes in the MAGMA analysis presented in the supplement from the recent largest GWAS study (Demontis et al., *Nature Genetics* 2023). We have modified our discussion to include this with the following statement:

“Although there was no overlap with the 76 prioritized risk genes by positional and functional annotation or the 45 exome-wide significant genes identified in the recent large ADHD GWAS⁴, there was overlap between genes mapped from externalizing-related disorders more broadly⁴².”